# Rock Sanctuaries, Sacred Landscapes, and the Making of the Iberian Pantheon

Alejandro G. Sinner [1,*] and Joan Ferrer i Jané [2]

1   Greek and Roman Studies, University of Victoria, Victoria, BC V8P 5C2, Canada
2   LITTERA Group, University of Barcelona, 08001 Barcelona, Spain
*   Correspondence: agsinner@uvic.ca

**Abstract:** Sanctuaries are common spaces of interaction between humankind and the gods. In many religious systems, mountains and other elevated topographical features are known to have formed part of these privileged spaces of communication. It is not surprising that open-air and, in many cases, rock sanctuaries are the cultic spaces par excellence among the pre-Roman peoples of the Iberian Peninsula. In this article, we offer a more nuanced picture of these architectonically humble but culturally rich sacred spaces by studying the Palaeohispanic inscriptions recorded in rock sanctuaries located in the territories of the Iberian peoples (fourth–first centuries BCE). Special attention will be paid to the corpus of inscriptions in Cerdanya (Pyrénées-Orientales and Catalonia), where more than 150 texts have so far been identified. After a brief introduction contextualizing the Rock Sanctuaries, the Iberian language, and the epigraphic habit of its speakers, the first section of our article analyses the characteristics that enable us to interpret most of these inscriptions as religious and votive formulations. The second half of the paper discusses what these inscriptions can reveal about the Iberian pantheon and how these rock sanctuaries formed a consolidated religious landscape that was to survive the Roman conquest. The reinterpretation of the Celtiberian sanctuary of Peñalba de Villastar will be fundamental to put forward the hypothesis that, while Iberian and Celtiberian places of worship and pantheons had points of contact, they were mostly dissociated from each other prior to the Roman arrival.

**Keywords:** Iberian language; religion; deities; Iron Age; Hispania; Palaeohispanic epigraphy; pantheon

## 1. Introduction: Rock Sanctuaries, Iberian Language, Epigraphic Habit, and Religious Texts

Rock sanctuaries are the cultic spaces par excellence among the pre-Roman peoples of the Iberian Peninsula. Are places adapted to the topography of the terrain, characterized by a location associated with particular natural conditions, the absence of constructions inspired by Hellenic architectural models and a variety of epigraphic practices that, in many cases, employed Palaeohispanic scripts. Rock sanctuaries would not exist—not as we know them—without graffiti. Inscriptions are the core element that redefines the space. In some cases, what creates the monument (e.g., Ragazzoli 2018), and the only remaining traces of the events, visitors, and religious exchanges that once occurred. Several authors have consistently proved that graffiti can participate in the logic of cultural appropriation, transforming through the act of writing public spaces and everyday objects for the benefit of private individuals or communities (e.g., Olton 2018). In the case of rock sanctuaries, the act of writing can become the main channel of communication between one individual (or a collective) and the deity, helping them to establish a divine identity (Beard 1991). On the one hand, the combination of the above characteristics enabled the space for movement and ritual to be organized; on the other, new meanings that were associated with cultic practice and specific deities could be generated. As we shall see, the consolidation of rock

sanctuaries was fundamental in the construction of shared practices and rituals among the Iberian peoples and helped them to develop a collective memory and in the formation of sacred and symbolic landscapes.

The Iberian language is attested in over 2200 inscriptions across a wide geographical area, from Hérault (France), coexisting with the Gallic language, to Jaén (Andalusia, Spain). Nevertheless, Iberian remains essentially undeciphered, and our knowledge of the language is limited. There are only a few bilingual inscriptions, and no other language close to Iberian has survived, which is a drawback that makes it impossible to understand well the texts written in Iberian. Nevertheless, our understanding of the Iberian language is advancing thanks to internal analysis of the inscriptions. Identifying repetitive patterns together with the support of contemporary epigraphic parallels makes it possible to understand the meaning of the shorter words (on the Iberian language see: Moncunill and Velaza 2017, 2019; Velaza 2019a with bibliography). Some nouns usually appear on the same type of objects, such as **baikar**[1] on small ceramic cups or **śalir** on silver coins. Few verbal forms are recognizable; **eŕoke**, for example, is widely used on lead sheets, while **egiar** probably has its Latin equivalent in *fecit*. Thanks to a bilingual text from Tarragona, it is also considered plausible that the form **aŕe** is the equivalent of the Latin *hic situs/a est*. The Latin equivalent of *filius*, **eban**, could have been identified as well, although it could also be the equivalent of the Latin *coeravit*. Personal names are the best-known category in Iberian since some Latin inscriptions, including Iberian anthroponyms, exist (e.g., TS = *CIL* I 709). Thanks to them, we know that most Iberian names are composed of two elements that are frequently combined with each other. When accompanying personal names, morphemes are identified acting as case marks: genitive (**ar/en**), dative (**e**), or an ergative mark, which would indicate the agent of the action (**te**).

A serious limitation in the study of religious and/or sacred texts, however, is that deity names, expected in this type of formulation, follow the same structure as personal names, which complicates differentiating the name of a god from that of an individual. The situation becomes even more confused if we consider that the worshipper and the worshipped tend to appear together in many inscriptions, obscuring who is who. To respond to this conundrum, in this paper, several methodological approaches are proposed.

The chronology of Iberian inscriptions, key for dating the texts to be discussed in this article, covers from the fifth century BCE until approximately the end of the reign of Augustus. Although the origin of the Iberian script, like all the Palaeohispanic scripts, is solely related to the Phoenician script (see: Ferrer i Jané 2017a with bibliography), the acquisition of the practice of writing by the northern Iberians, probably, is a direct consequence of their contact with the Greek colonies in southern France and north-eastern Spain. The earliest securely dated inscriptions are graffiti on Attic vessels (fifth and fourth centuries BCE) found in the excavations carried out at the oppidum of Ullastret (Girona) in north-eastern Spain near the Greek colony of Emporion and from Pontós (Girona) (Ferrer i Jané et al. 2016). From this same period, Greek inscriptions made on lead sheets (e.g., Pech Maho or Emporion; Rodríguez 1996; Santiago 2003 with bibliography) document the participation of Iberians in trade agreements. Therefore, it is accepted that most of the lead sheets with Iberian inscriptions are also trade-related. It is established, however, that Iberian inscriptions that can be chronologically ascribed to dates between the fifth and third centuries BCE do not include funerary texts. The known cemeteries exhibit iconographic forms of ancient tradition or oriental influence (via southern Spain) and, more than anything else, anepigraphic funerary steles (on Iberian funerary epigraphy, see Velaza 2017, 2018).

It is only between the fourth and the third century BCE that the epigraphic habit seems to have been extended into the religious sphere. As will be seen, the use of writing, gradually adopted in the rock-face sanctuaries along the Mediterranean coast, became a key element in Iberian ritual and cultic practices. Later in the third century BCE, painted Iberian pottery was produced, and, in some of these vessels, the ritual was mixed with ethnic and elite self-representation, connecting the two worlds. A good example of possible

intertextuality between sacred inscriptions on two different surfaces and materials is the *kalathos* from Llíria and the inscriptions in the Tarragón shelter (see below).

Nevertheless, it was during the second century BCE, as a consequence of the Roman presence, that constant and rapid changes occurred in Iberian epigraphic culture, and religious epigraphy was not an exception. With the influence of Roman public writing models, new categories of epigraphy and inscriptions on different materials appear, as well as the development of carving techniques and the use of new palaeographic forms and formulae. At that moment, tomb inscriptions started to incorporate epigraphic texts as part of their design. Initially, the textual component complimented the existing iconographic funerary language, as can be seen in the funerary monument at Tamarit de Llitera (HU.01.02) and the stele from El Acampador (BDH Z.16.01). In the latter, text and iconography share the space. The upper half is sculpted in the form of a feline, four motifs interpreted as a *scutum*, and three *caetrae* are carved underneath, while the text occupies a large space in the lower half of the monument. Subsequently, a gradual disappearance of decoration in favour of textual predominance was to take place, culminating in semi-circular steles, such as the one from Sinarcas (F.14.1), where text is the only element present (Velaza 2018, pp. 173–174). Parallel and related to this phenomenon, rock inscriptions, until then mainly associated with rock faces, developed into sanctuary epigraphy at a time when sacred spaces were monumentalized. Perhaps the best example of these practices can be seen at the sanctuary of Liber Pater at Muntanya Frontera (Nicolau 1998; Simon 2012). A form of Iberian epigraphy that can be described as public and applicable to the religious sphere emerged before the Iberian language rapidly fell out of use towards the end of the first century BCE.

## 2. Materials: Inscriptions in Iberian Rock Sanctuaries

Since the publication of *MLH* III (1990), which included only two inscriptions from rock sanctuaries, there has been a substantial number of discoveries, accompanied by the requisite scholarly analysis, which has increased the corpus of inscriptions to a current total of 53 rocks. Most of them are concentrated in the Cerdanya region, administratively divided between the department of Pyrénées-Orientales (France) and the province of Girona (Spain) (Figure 1). In this area, 45 rocks with more than 175 texts have been identified and, in many cases, studied and published (Campmajó and Ferrer i Jané 2010; Ferrer i Jané 2010, 2012b, 2013a, 2014a, 2015b, 2015c, 2016, 2017b, 2018a, 2018b, 2019, 2020). To the above, we must add four rocks from the Catalan region of Osona (Barcelona): Les Graus in Masies de Roda de Ter (*MLH* III, D.3.1); two from L'Esquirol and one from Sant Martí de Centelles (Ferrer i Jané 2014b, 2021). In the province of Lleida, we must mention the inscription on the Roca dels Moros (El Cogul; *MLH* III, D.8.1). Moving south, in Valencia, two more rock inscriptions are known: the Burgal shelter in Siete Aguas (Fletcher and Silgo 1996) and the Tarragón shelter in Losa del Obispo (Ferrer i Jané 2018c). Finally, in Albacete, two additional inscriptions complete the corpus: La Camareta Cave in Hellín (Luján and López 2016) and the Reina shelter in Alcalá del Júcar (Ferrer i Jané and Avilés 2016). These last two texts, however, unlike the previous ones, which used the north-eastern Iberian script, use the south-eastern Iberian script (on the Iberian writing systems see Ferrer i Jané and Moncunill 2019, pp. 82–101; Velaza 2019a, pp. 160–97). Additionally, it is worth pointing out that there is a record of an inscription from Badalona (Barcelona), supposedly written in the Iberian script, destroyed by modern quarrying work before it could be removed and studied. Another votive inscription, in this case in Latin and dedicated to *Soli Deo* (CIL II 4604), was saved from the same location.

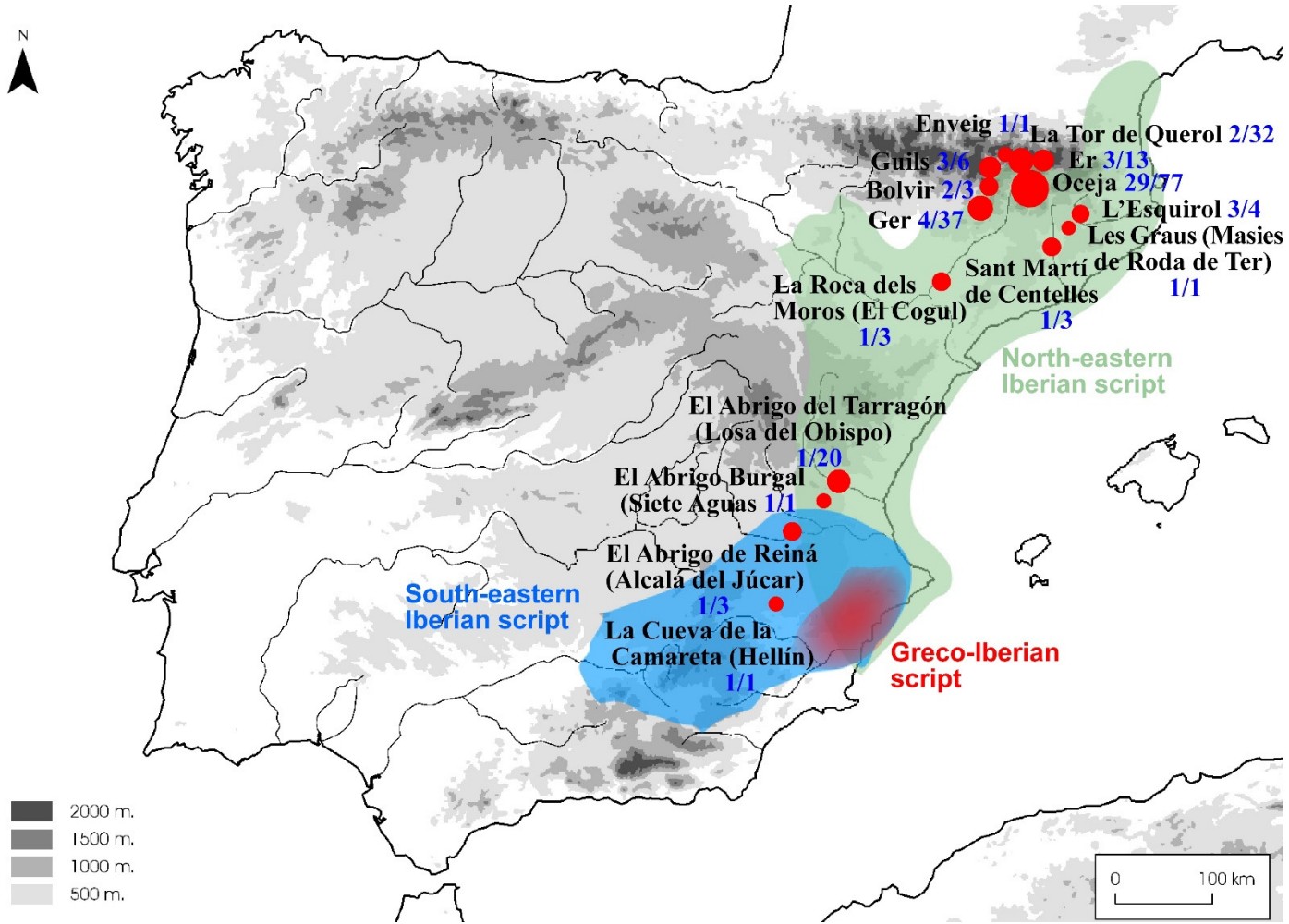

**Figure 1.** Iberian rock inscriptions included in this study. The first number indicates the number of rocks and the second, the number of inscriptions.

### 2.1. Characteristics and Chronology

It is quite difficult to date this body of inscriptions with any degree of certainty. Therefore, a relative chronology can only be put forward on the basis of palaeographic analysis. The inscriptions on the Mediterranean coast, usually the ones considered the oldest (fifth–third centuries BCE), present heterogeneous shapes and are characterized by the simultaneous presence of simple and complex variants of the Iberian script that is known as dual writing (Ferrer i Jané 2005; Ferrer i Jané and Moncunill 2019). On the other hand, the inscriptions considered to date from the second–first centuries BCE are usually much more homogeneous and are characterized by only displaying one of the two script variants, usually the simple one (not dual).

Applying the above criteria to rock inscriptions, it is possible to suggest an approximate date because the non-dual writing system was used for most coin legends (Ferrer i Jané 2012a; Ripollès and Sinner 2019) and many other inscriptions that can be securely dated to the second–first centuries BCE. The corpus of rock inscriptions is normally divided between those texts using the dual and non-dual writing systems. In the cases where the sample preserved is large enough, the division between these two categories (dual and non-dual inscriptions) seems to be relatively balanced. Thus, just over half of the inscriptions in Cerdanya are dual (Figure 2: 1, 2, 6, 7, 9, 10, 14, 15, 17, 19, and 22), while the rest are not (Figure 2: 3, 4, 5, 8, 11, 12, 13, 16, 18, 20, and 21). In Osona, where only four inscriptions are known, the division is similar: the texts from Les Graus and Sant Martí de Centelles are dual, while the two inscriptions from L'Esquirol (Figure 3) are not. Finally,

the script used on La Roca dels Moros is not dual either, but the ones used in the Tarragón shelter and Burgal shelter are dual, perhaps even of the extended dual type (Ferrer i Jané 2013b, 2015a).

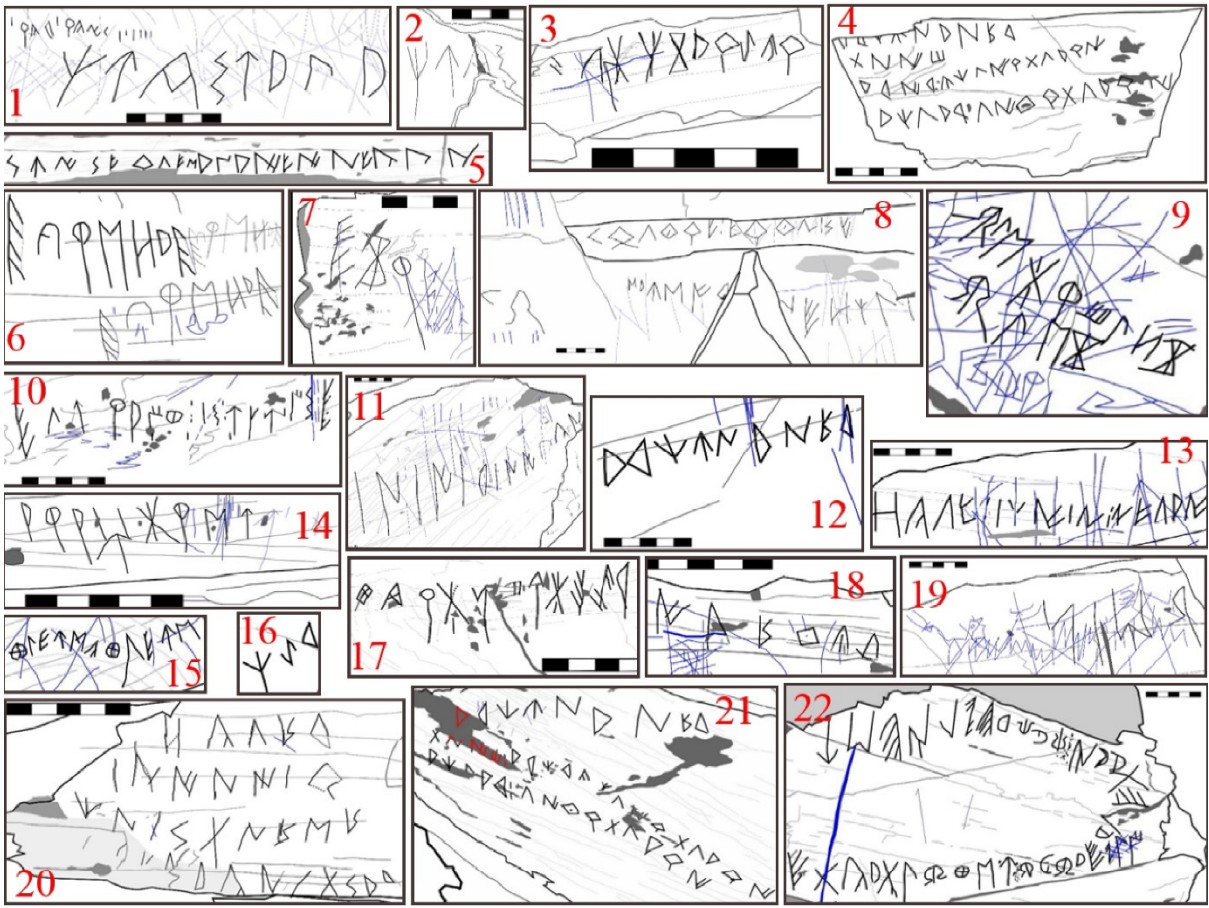

**Figure 2.** Iberian rock inscriptions from Cerdanya.

Regarding their typology, the supporting rocks can also be divided into two main groups: the ones in Cerdanya are usually free-standing schists and differ from those in the remaining areas, which are calcareous rocks in the form of shelters (cave-like structures). This division does not seem to serve any functional purpose and is most likely to be the consequence of geological and anthropological causes (e.g., hardness of stone, quarrying activities), which clearly explains why only inscriptions in unpopulated areas have survived. Human intervention and erosion probably caused the destruction of many inscriptions engraved in less well-protected areas.

Paleographically, the dimensions of the signs vary, but in no case are they large enough to consider that they were intended to be seen from a distance. This is important because it excludes the interpretation of these texts as territorial delimitations or as having been for public and monumental purposes. The largest signs do not exceed 10 cm. However, it is possible to establish a different pattern between the rocks of Cerdanya and the rest. In the former area, the height of the signs varies between 0.5 and 3.5 cm on average, while in the case of the remaining rocks, the height is considerably larger, ranging between 3 and 10 cm. The surface where the inscriptions were engraved could help to explain this tendency. On calcareous rocks, the production of inscriptions is much harder than on schists, where the writing of limited dimensions is easy to produce. It should be noted that the time elapsed since they were made and the patina accumulated on the surface of the rock makes their identification difficult. However, at the time of their execution, it would have been easier to identify and read them thanks to the contrast of clear lines without any patina on an

otherwise dark surface. In any case, many inscriptions are formed by signs with heights of less than 1 cm, compatible with a type of register that seems more personal than public in character.

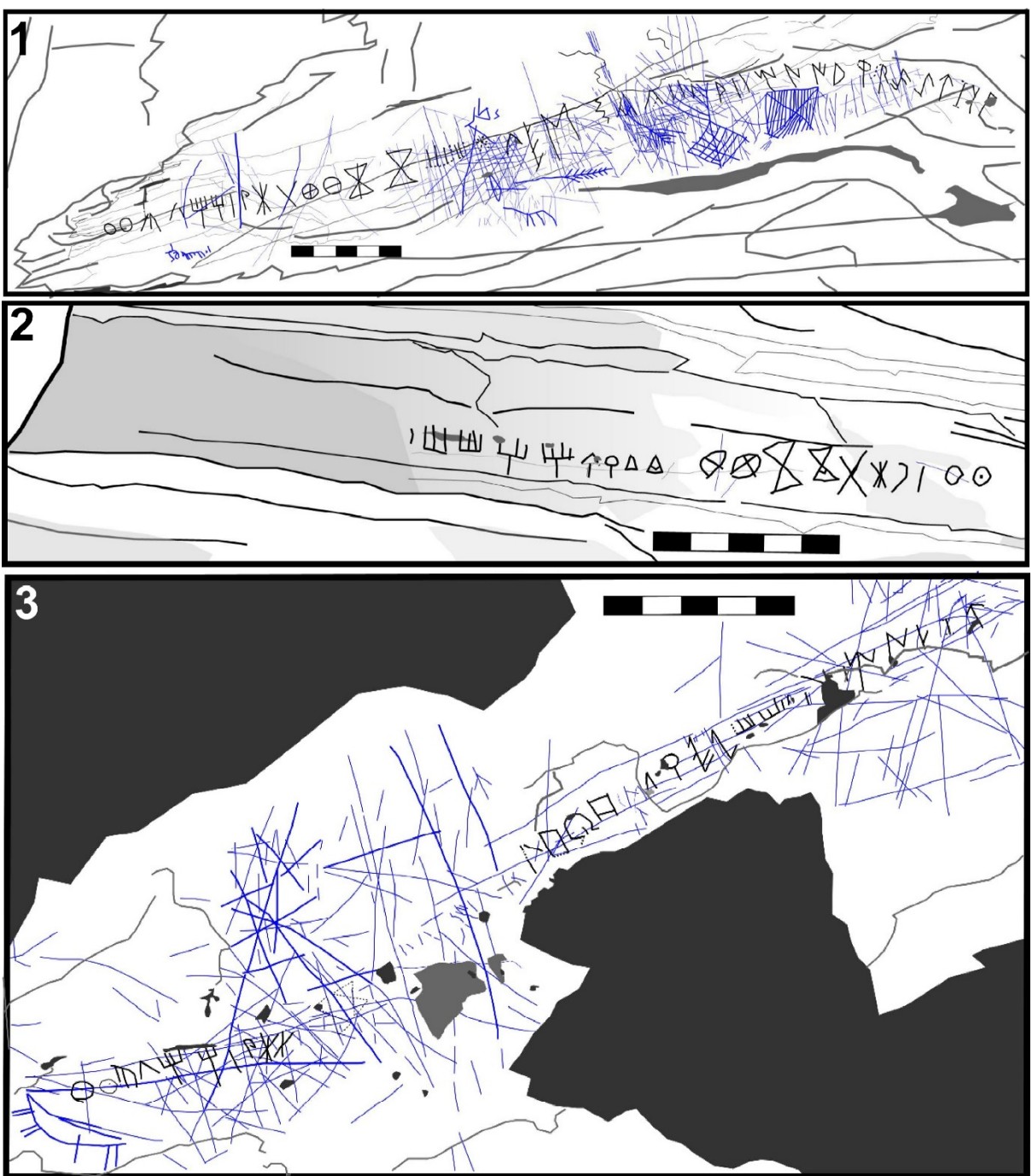

**Figure 3.** Iberian abecedaries recorded in Cerdanya rock-sanctuaries. 1. Ger. 2. Bolvir. 3. La Tor de Querol.

### 2.2. Votive and Religious Inscriptions

Today, the religious and votive nature of these inscriptions, especially in the case of Cerdanya, is broadly accepted (De Hoz 1995, p. 14; Rodríguez Ramos 2005, p. 66; Ferrer i Jané 2010, p. 58; Velaza 2014; Sinner and Velaza 2018, p. 5). In the following lines, the main arguments in favour of this interpretation are presented, but we must remember that the

language remains largely undeciphered and, therefore, few facts can be taken for granted without further discussion.

An initial persuasive argument to justify the religious nature of these inscriptions relies on the fact that some of the Iberian texts in rock sanctuaries shared the space with Latin ones that present no doubts as regards their interpretation. The clearest case is that of El Cogul, where the text *Secvndio/Votvm Fecit* appears next to the Iberian ones (*MLH* III: D.9.1). It can also be the case of a Latin inscription from Er (Cerdanya) (Ferrer i Jané et al. 2020). The religious meaning of the inscriptions and symbols traced on many of the rocks studied in this study in later periods—from the Christian era in general, but also from the Arabic texts recorded from La Camareta (Hellín, Albacete)—also points towards the religious nature of these places and their value in the construction of sacred landscapes and religious memory.

Another argument favouring the religiosity of these inscriptions is the engraving of Iberian abecedaries. In line with Greek, Latin, and Etruscan parallels, abecedaries are a characteristic feature of sacred landscapes and not always a practice related to the learning of writing (on performative and symbolic writing, Beard 1991; Hall 2000; Velaza 2012). Abecedaries are common in our corpus of rock inscriptions (Table 1) in both non-dual (L'Esquirol, and La Tor de Querol) and dual writing (Ger, La Tor de Querol, and Bolvir, Figure 3), showing the survival of this practice over the course of time. The initial sequence of the dual abecedary of La Tor de Querol, **kugutudutidibabitada**, identifies the first characteristic sequence of the dual Iberian abecedary and coincides with the inscription in the abecedary at Ger. The non-dual abecedary of La Tor de Querol, while only partially preserved, is recognizable as such because the initial sequence **kutukiŕbitatiko** coincides with the initial sequence at L'Esquirol. Another possible non-dual abecedary was recently published in an inscription from Bolvir; however, if considered an abecedary, we cannot be certain it would be an incomplete one.

**Table 1.** Iberian abecedaries recorded in rock sanctuaries.

| Location | Text | Typology |
|---|---|---|
| Ger | *kugu<u>tudu</u>tidibabitadatedekogotodo++eśskaga++a+mniŕбekigiuI<i>ḿ</i>+* | Dual Std. |
| Bolvir | *kugubabitadakogotede+tuduźutiditodo+[* | Dual Std. |
| La Tor de Querol | *kugutudutidibabitada++<u>kogo</u>[c5/7]++bubeo++ŕkigi<u>todo</u>[+]+mnḿIu* | Dual Std. |
| L'Esquirol | <u>kutukiŕ</u>bitatikokabastokeaubooelInḿite[-]/śr<u>be</u>[+] | Non-dual |
| La Tor de Querol | **kutukiŕbitatiko([)** | Non-dual |
| Bolvir | **K<u>utu</u>bitatiko** | Non-dual |

A new inscription from Ger containing the formula **neitin iunstir**, probably a propitiatory or salutation form since it usually opens the texts where it appears (Ferrer i Jané 2016 with bibliography), reinforces the votive nature of the Iberian rock inscriptions. For the first time, this new inscription simultaneously includes the four elements—**neitin**, **iunstir**, **baŕbin,** and **uskei**—completely, a structure that until now had only been recorded in fragmentary form since these elements always appear in pairs or groups of three in the other known inscriptions. Particularly interesting for this interpretation are the three lead sheets with a possible cultic function from the tomb of La Punta d'Orlell (Vall d'Uixó) (*MLH* III: F.9.5–7; Ferrer i Jané 2016, p. 25). Therefore, all the parallels seem to point towards interpreting this inscription as a votive, **neitin** being the name that would identify the divinity, while **iunstir** could be a verb that would indicate the propitiatory action. Likewise, the formulae from Oceja (Table 2) also support the religious interpretation.

**Table 2.** Formulae recorded in the text from Oceja.

| Text | | | | | | | |
|---|---|---|---|---|---|---|---|
| **artiunan er/** | **tanito/** | **arir** | **kati** | **li** | | **ŕ** | **talaŕi/** |
| | | | **atilar** | **li** | **ku** | **ŕ** | **talaŕi** |
| **artiunan er/** | **ta<u>nito</u> ·** | **artir** | **kati** | **li** | | **ŕ** | **talaŕi/** |
| | | | **atilar** | **·** | **li** | **ku** | **ŕ** | **talaŕi** |
| **artiunan er** | | | | | | | |

Repetitive elements also favor the votive interpretation of rock inscriptions. The clearest case is that of **urdal**, which appears in the Tarragón shelter nine times. Everything seems to point to **urdal** being a divinity. In the first place, its structure does not respond to the usual two-member structure characteristic of Iberian personal names; nor do any of the most common Iberian personal name formants appear. A strong argument for rejecting the nature of **urdal** as a personal name is palaeographic. All occurrences of **urdal** in the same inscription could be expected to be identical from the calligraphic perspective since the same individual, **urdal**, could have made them. While the signs **u**, **r**, **da,** and **l** are relatively similar and do not exhibit great variation, after a detailed in situ inspection of the texts, they do not give the impression that all the **urdal** forms were made by the same hand. We also cannot trace other anthroponyms among the rest of the inscriptions in the Tarragón shelter. Finally, the strongest argument towards understanding **urdal** as a deity is its possible relation with the Basque deity *Vrde*, identified in a Latin inscription from Muzqui (Navarra) (Velaza 2012; Ferrer i Jané 2018c, p. 237).

Finally, exceptional elements can be detected among the rock inscriptions. Excellent examples are the two radial inscriptions from the Tarragón shelter (Figure 4) (Ferrer i Jané 2018c, p. 238; Rodríguez Ramos 2020, p. 271). The only parallel to these radial representations in the Iberian corpus is the text painted on a ceramic vessel from Llíria (*MLH* III: F.13.3; Ferrer i Jané 2018c, p. 245). Although none of the three texts are identical, the similarities are evident, a circumstance that allows us to think that, at least, they represent a similar, if not an identical, concept. The radial disposition also seems significant for interpreting these texts as religious. Among the many *post cocturam* Iberian graffiti known and interpreted as property marks (personal names), none replicates this radial structure. Although the reading of all the signs is not completely clear, the main difference between the two radial inscriptions in Tarragón is the presence in one of them of an additional sign **ŕ**, which turns the text into **kauŕgo** while the other reads **kaugo**. The differences between the rock inscriptions in Tarragón and the painted inscription are more pronounced. In the ceramic inscription, the missing sign is the sign **u**, so the reading is **kaŕko**. Additionally, it also presents two different signs in another section of the inscription. While the two inscriptions from Tarragón contain the element **beřolé**, in the graffiti from Llíria, it appears as **elolé**. Based on what we know of the Iberian language, these two elements could be the same since the dropping of the labial, and the replacement of **ŕ** by **l** are well-documented phenomena (Quintanilla 1998, p. 254).

The closest parallel we can suggest for the element **kauŕgo/kaugo/kaŕko** is the element **kaukoŕ**, which could be a fourth variant of the same word. The latter has been identified in an incomplete inscription on the rim of a *kalathos* from the important Iberian sanctuary at Muntanya Frontera (Sagunto) (*MLH* III: F.11.32). It must be said that, based on the position in which **kaukoŕ** appears in this inscription, it could be acting as a verb (Velaza 2008, p. 302) rather than as a theonym (Rodríguez Ramos 2014, p. 213). However, taking into account the location where the inscription was recovered, the doubts in the reading of the final sign, and its fragmentary state, perhaps the element identified was strictly **kauko**, as in one of the inscriptions from Tarragón. In addition, because of its compatibility with the structure used in Iberian onomastics and in the light of the two radial inscriptions mentioned above, we believe the verbal function should be replaced in favour of a deity name. The cult of Liber Pater is well documented at the sanctuary of Muntanya Frontera in

Roman times. Prior to the Roman arrival, it could have been dedicated to a local divinity, perhaps the same one that is hidden under the forms **kauŕgo**, **kaugo**, **kaŕko,** and **kauko(ŕ)**.

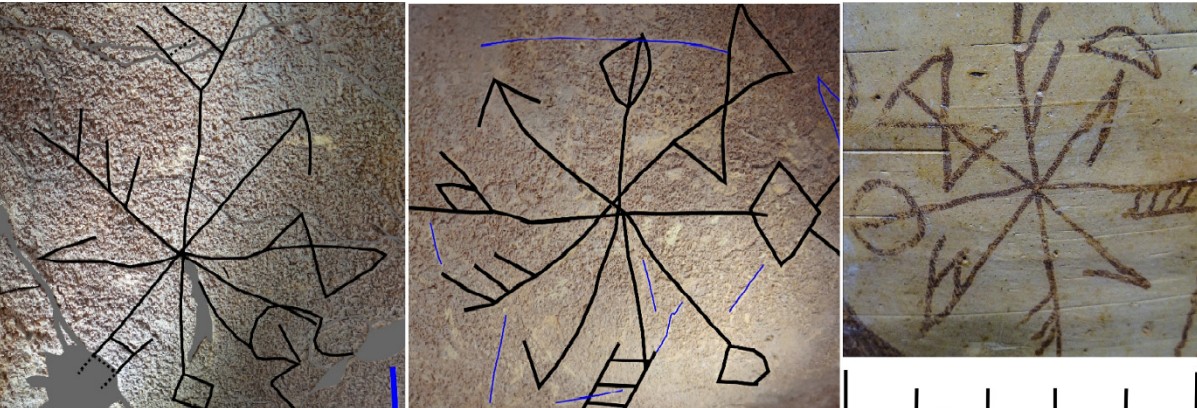

**Figure 4.** Radial inscriptions: Tarragón 12, Tarragón 13, and F.13.3.

## 3. Methods and Results: Identifying Iberian Deities

The list of Iberian deities is very short. In the classical sources, only Macrobius (*Saturnalia* 1.19.5) cites *Neton*(Granada) as a deity, which could be related to the Iberian **neitin** (Beltrán 1970, p. 521; Almagro-Gorbea 2002; Silgo 2004, p. 196; Corzo et al. 2007, p. 255; Orduña 2009, p. 507). Furthermore, very few names of Iberian deities can be identified with certainty in Latin inscriptions, and their Iberian nature is not always clear. The clearest examples always follow the same two-member structure, *Betatvn*/**bete** + **atun** (Corzo et al. 2007; Ferrer i Jané et al. 2018, p. 182), *Salaeco*/**śalai** + **ko** (Velaza 2015), *Sertundo*/**seŕtun** + **do** (Vidal 2016) and *Salagin*/**sal(a)** + **(a)gin** (Gimeno and Velaza 2021), which, without a clear context, can easily be confused as personal names. Thus, it is not an easy task to distinguish theonyms from anthroponyms on the basis of their structure alone.

Therefore, the texts presented above represent the best corpus of data available to isolate and identify the very elusive Iberian deities. Due to the nature of votive texts, the names of the dedicants and other usual elements, such as the dedication verb and the dedicated object, and the name of the divinity should appear in at least some of the texts. If this hypothesis is accepted, the repetition and position of certain names in these inscriptions could be a good argument to suggest them as deities. A good strategy to distinguish the theonyms from the anthroponyms could be to identify the morph(s) that are only combined with them. It makes sense that in a very limited context such as rock inscriptions, the behaviour of personal names and deities should be different, with a distinct syntactic position in each case marked by the corresponding morph (Table 3).

One of the most frequent morphs recorded in the rock inscriptions is **e**. It has been suggested that **e** is the dative mark in the Iberian language (cf. Ferrer i Jané 2006, annex 2; Rodríguez Ramos 2017; cf. Moncunill and Velaza 2019, p. 236), a function that would correspond well with a theonym to which an offering is given. It appears in the inscription at Er accompanying **kebelkuŕ** (Figure 2: 8) (Ferrer i Jané 2015c, p. 11, no. 3) and **eŕkunbas** (Figure 2: 8) (Ferrer i Jané 2015c, p. 11, no. 3), and in Oceja accompanying *egeŕśor* (Figure 2: 6) (Ferrer i Jané 2015c, p. 11, no. 3), up to three times. The same morph also probably appears in **okal** (Figure 2: 13) and in **[a]dinbastaneś** (Figure 2: 20) (Ferrer i Jané 2017b, p. 12, no. 31), both in Oceja as well. On the same surface at La Tor de Querol **tikanal** (Figure 5) is documented three times and **balkar** twice (Figure 5), all five cases followed by the morph **e**. Less clear is the case of **garde** at Ger (Figure 5), where the morph **e** might also be visible (Ferrer i Jané 2019).

**Table 3.** Morphs discussed in this article and their use in inscriptions.

| | Morph | Name | | | | |
|---|---|---|---|---|---|---|
| **DEITY NAMES?** | e | **okal** (2) | **kebelkuŕ** | **eŕkunbas** | *egeŕśor* (3) | **[a]dinbastaneś** |
| | e | *tikanal* (3) | *balkar* (2) | *gard(e)*? (4) | **unibas** | |
| | i? | *oskikiŕ*? (2) | | | | |
| | er | **okal** | *egibal* | **artiunan** (3) | *idaŕ* | *uńmis* |
| | ir | **śauś** | **gais** (2) | | | |
| | ka | *teleuś* | **anaieine** | | | |
| | ika | *taŕ* | | | | |
| | ike | **baŕka** | **urdal** | | | |
| | kate | *akietau<u>kemtaŕ</u>* | <u>*balśiriste*</u> | | | |
| | śu | *aŕamtaŕ* | *edagardal***(bete)** | | | |
| | - | *teleuś* | *diukas* (2)/*tiuga* | *urdal* (9) | **banbaibar** | *garde*? (4) |
| | - | **śauś** | **kebe** | *balkar* (2) | **baŕkar** (2) | *kau(ŕ)go* (2) |
| **UNCERTAIN** | - | **urbir** | *amban* | *ekoŕ* | **aŕgiuŕ** | |
| | | *tikanbiuŕ* | *ekeŕbeleś*‿ | *begeber* (2) | | |
| | - | **suisebeleś**‿ | *tartiar*? | *kobeśir* | | |
| | - | *belśtaŕ* | *toloko* | *belśko* | ]*skon* | |
| **PERSONAL NAMES?** | | *oŕdinkali* | | | | |
| | te | **aiunildiŕ** | *uŕkesir* | *eluŕai* | | |
| | I | *oŕdinkali* | *tigirsadin* | | | |
| | ḿi | <u>ba</u>ŕ<u>ebilos</u> | | | | |
| | ar | *tarti*? | | | | |

Also quite common is the morph **er**, probably a dative mark (Orduña 2005a, p. 228; Ferrer i Jané 2018a, p. 118; Rodríguez Ramos 2017, p. 119), which combines with *egibal*, *idaŕ*, **okal**, and **artiunan** (Figure 2: 4, 12 and 21) (Ferrer i Jané 2010, p. 258, no. 9; 2010, p. 268, no. 28; 2015c, p. 17, no. 14). The latter form appears three times in Oceja and, together with *uńmis* (Figure 2: 10), in the text from Enveig (Ferrer i Jané 2015c, no. 18). The element **śauś** at Ger and **śausir** at Er (Figure 2: 8) could also fit in this group if the morph **ir** were a variant of the morph **er** (Untermann 1990, p. 165; Ferrer i Jané and Garcés 2013, p. 109; Rodríguez Ramos 2017, p. 141; Ferrer i Jané 2019, pp. 47 and 51). The same morph **ir** could follow *gais*, which appears twice in the form *gaisir* at the rock of Sant Martí de Centelles (Ferrer i Jané 2021), which was already known as a personal name formant, as in GAISCO (Ferrer i Jané et al. 2018, p. 182).

Another morph recorded is **ka** (cf. Ferrer i Jané 2006, annex 11; cf. Moncunill and Velaza 2019, p. 263), which has the variants **ke**, **ika,** and **ike** and accompanies personal names in accounting texts recording quantities. This is likely to indicate the destination of the transaction (Ferrer i Jané 2020). In rock inscriptions, **ka** appears next to *teleuś* (Figure 2, 15) in Oceja and perhaps next to **anaieine** at Guils (Figure 2: 5) (Campmajó and Ferrer i Jané 2010, p. 258, no. 16). The new rock inscription from Ger with the text **urdalike** (Ferrer i Jané 2020), contains the **urdal** nucleus and the morph **ike**, reinforcing the hypothesis that it is a variant of morph **ka**, and therefore confirming the hypothesis that in the Iberian language it had a similar function to that of the dative, probably the benefactor, also compatible with theonyms. In the same sense, the new reading of the rock inscription from L'Esquerda (Ferrer i Jané and Rocafiguera n.d.), *taŕika*, could be interpreted as a nucleus *taŕ* followed by the variant **ika**. The morph **kate**, perhaps a variant of **ka**, or a combination of this with the morph **te**, is documented in two cases accompanying complex elements of unclear segmentation: *akietaukemtaŕ* in Oceja (Figure 2: 17) and *balśiriste* from Reiná (Ferrer i Jané 2006, annex 2 with bibliography).

Another of the morphs identified is **śu**; although its function remains unclear, it accompanies *aŕamtaŕ* in Bolvir (Figure 2: 14) and perhaps *edagardalbete* (Figure 2: 22) (Campmajó and Ferrer i Jané 2010, p. 260, no. 21, p. 257, no. 5) in Oceja. Nevertheless, the most probable deity name is strictly *edagardal*.

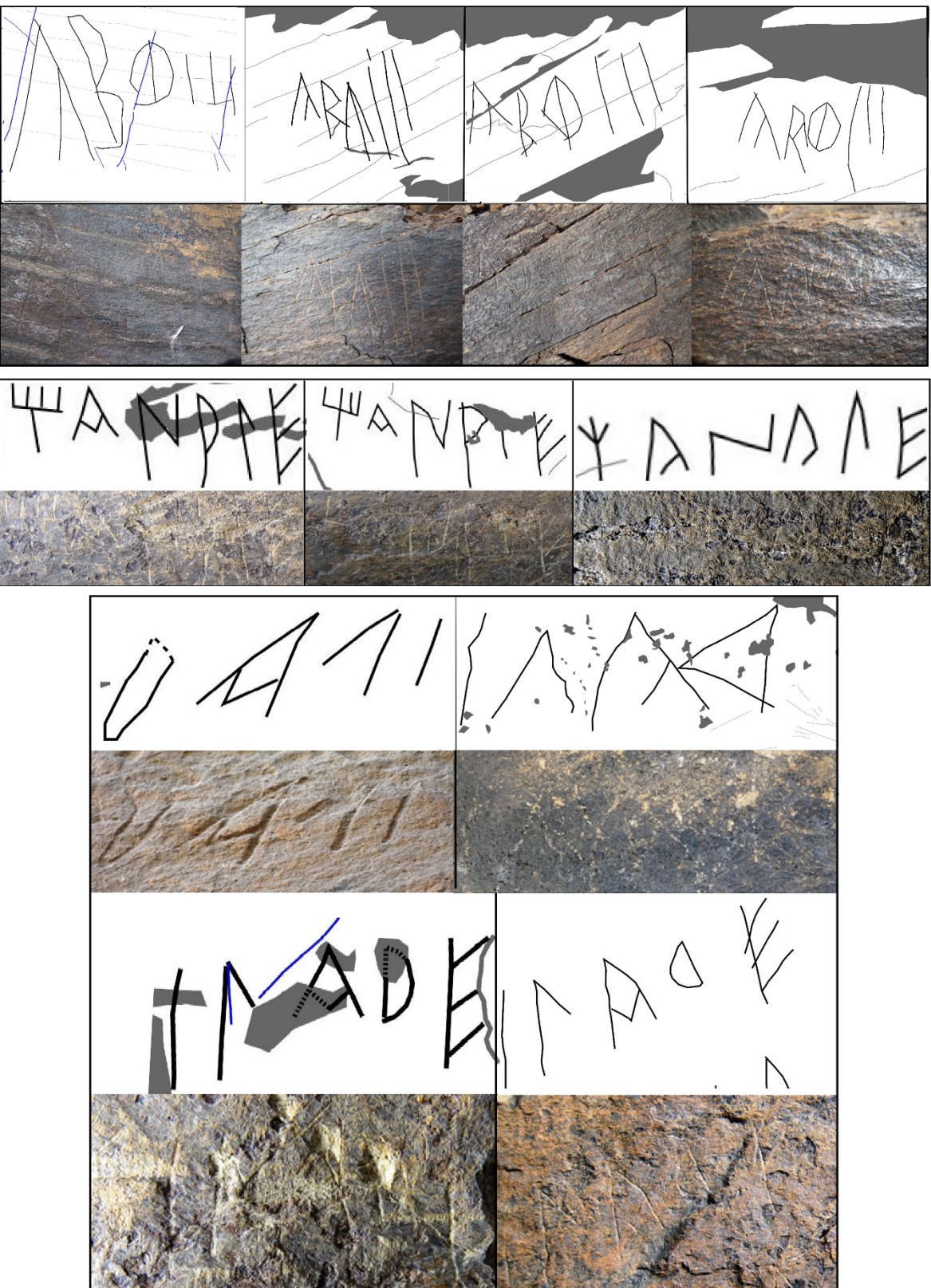

**Figure 5.** **Garde**, **tikanal**, and **balkar**.

　　The morph **te** (cf. Ferrer i Jané 2006, annex 2; cf. Moncunill and Velaza 2019, p. 213), which in other contexts usually accompanies personal names and marks the agent of the action, is identified in **aiunildiŕ** at El Cogul (D.8.1), in **uŕkesir** from Reiná and perhaps in

*eluŕai* from Enveig (Figure 2: 10) (Ferrer i Jané and Avilés 2016; Ferrer i Jané 2015c, p. 16, no. 18). The morph **I**, a sign that probably combines a nasal and a vowel value, such as **ḿ**, could be a variant of the more frequent **ḿi**, which usually accompanies personal names in inscriptions marking the property of an object. In the rock inscriptions, **I** is recorded in two contexts in Oceja combined with *oŕdinkali* and *tigirsadin* (Ferrer i Jané 2018a, p. 109). The morph **ḿi** can also be directly identified in an inscription at Guils accompanying the personal name **baŕebilos**. Finally, there is one case in which the morph **ar** could be used following **tarti** (Ferrer i Jané 2018a, p. 109), which is also characteristic of personal names and is usually interpreted as a genitive mark, although it could also correspond to the personal name **tartiar**, without a morph.

In other inscriptions, the onomastic elements do not present any morphs. That is the case of *belśtaŕ*, *toloko*, *belśko*, and ]*skon* in the texts from the rock sanctuary at Er (Figure 2: 9), of **suisebeleś** at Guils (Figure 2: 5), and *kobeśir* at La Camareta. The same principle applies to one of the best candidates to be a divinity, *urdal*, in the texts from the Tarragón shelter. Initially, neither of the two occurrences of *begeber* in Oceja (Figure 2, 22) has an associated morph if we identify the following element as *ekilie* and *ekele* (Campmajó and Ferrer i Jané 2010, p. 257, no. 2). Other onomastic elements without a morph in the rock inscriptions are **banbaibar** in Oceja (Figure 2: 11) (Ferrer i Jané 2015c, p. 16, no. 27), **ekeŕbeleś** at La Tor de Querol (Ferrer i Jané 2015c, p. 11, no. 2.) and *amban* (Figure 2, 19), *diukas* (Figure 2: 1 and 2) and *urbir* (Figure 2: 1) at Er (Ferrer i Jané 2015c, p. 11, nos. 3–4). It would also be the case of *tikanbiuŕ*, the corrected reading of a rock inscription from Oceja (Ferrer i Jané and Garcés 2013, p. 109). Among the recently published inscriptions, *bekoŕ* (Figure 2: 7) and **aŕgiuŕ** (Figure 2: 3) should be added to this list. In some cases, some of the elements discussed above and accompanied by one of the morphs are also documented without it, as in the case of **teleuś** (Figure 2: 15), **śauś** (Figure 2: 8), and **oŕdinkali**, both in the inscriptions from Oceja.

Many of the above elements also appear repeatedly in the corpus of rock inscriptions under study (Table 4). The above-mentioned element *urdal* is the most often recorded—appearing up to nine times on the same surface, and once at Ger (Cerdanya) (Ferrer i Jané 2020). Similarly, *balkar*/**baŕkar** appears seven times: the variant **balkar** twice on the same surface, and once at Sant Martí de Centelles and at El Cogul (Figure 5), in both cases without any morph, while the variant **baŕka(r)** appears twice at Er (Figure 2: 1), and once at Les Graus. This geographical distribution makes these two deities the first supra-local examples.

The element *garde* appears four times, three times on the same rock surface at Ger as well as on a different rock in the same location (Figure 5). Similar is the case of *tikanal*, which is recorded three times on the same rock surface at La Tor de Querol (Figure 5), like *egeŕśor*, which appears three times (Figure 2: 6) on the same surface at Oceja, while **okal** and **artiunan** are repeated three times on different surfaces also at Oceja (Figure 2: 12 and 21). The elements that are repeated twice are *teleuś* in Oceja (Figure 2: 15)—on the same rock surface—and *diukas* at Er, also on the same surface (Figure 2: 1 and 2), and perhaps **kebe**/**kebelkuŕ** also at Er (Figure 2: 8) if the first one is the simplification of the second.

The argument of repetition to support the interpretation of a word as a theonym has already been used in the case of the Celtiberian rock inscriptions of Peñalba de Villastar regarding the elements *Tvros* and *Calaitos*, but this is still contested due to some scholars considering them personal names (Marco and Alfayé 2008, p. 514).

The analysis of the corpus of Aquitanian inscriptions (Gorrochategui 1984; Gorrochategui and Sádaba 2013. See also Eduardo Orduña Database http://eorduna.awardspace.info/ (accessed on 7 June 2021)) can help to clarify the probability that a repeated element is a theonym. The Aquitanian corpus includes around 450 inscriptions, combining votive (two-thirds) and funerary (one-third) texts. Up to 65 elements are repeated in the total, but this number can be reduced to 47 if only the votive texts are considered. Even in the most unfavourable case, which considers—as happens with the Aquitanian corpus—that not all the inscriptions are votive in nature, of the ten most commonly repeated Aquitanian

onomastics (over five times), seven are deities, namely *Erriappo*, *Erge*, *Leherenno*, *Abelionni*, *Ilunni*, *Artehe*, and *Ageioni*, and only three are personal names, namely *Andos*, *Sembi*, and *Silex*. It is important to bear in mind that the latter is formed by one single element, and that facilitates its repetition. In contrast, the most common structure of Iberian personal names consists of two elements, and therefore the probability of repetition is lower than in the Aquitanian case (Ferrer i Jané 2019).

**Table 4.** Possible Iberian deities discussed in this text with their number of appearances in the corpus of rock inscriptions and the morph with which they combine.

| Site | Rep. | Location | Deity? | Morph |
|---|---|---|---|---|
| Tarragón (Valencia)/ Ger (Cerdanya) | 10 | Different Sites (Valencia, Cerdanya) | *urdal* *udal* *urdal* | *ike* |
| La Tor (Cerdanya), Er (Cerdanya), El Cogul (Lleida), Les Graus (Osona), Sant Martí de Centelles (Osona) | 7 | Different Sites (Cerdanya, Lleida Osona) | *balkar (2)* *balkar (2)* *baŕka(r) (1)* *baŕkar (2)* | *e* *ike* |
| Ger (Cerdanya) | 4 | 2 different rocks | *gard(e)* | *e (III)* |
| Oceja (Cerdanya) | 3 | 3 different rocks | **artiunan** | *er* |
| Oceja (Cerdanya) | 3 | Same surface | *egeŕśor* | *e* |
| Oceja (Cerdanya) | 3 | 3 different rocks | **okal** **okal** | *er* *e (III)* |
| La Tor de Querol (Cerdanya) | 3 | Same surface | *tikanal* | *e* |
| Er (Cerdanya)/La Tor de Querol (Cerdanya) | 3 | 3 different rocks | **diukas (2)/***tiuka* | |
| Tarragón (Valencia) | 2 | Same surface | *kauŕgo/kaugo* | |
| Oceja (Cerdanya) | 2 | Same surface | *teleuś* *teleuś* | *ka* |
| Er (Cerdanya) | 2 | Same surface | **kebelkuŕ** **kebe** | *e* |
| Ger/Er (Cerdanya) | 2 | 2 different rocks | *śauś* *śauś* | *ir* |
| Sant Martí de Centelles (Osona) | 2 | Same surface | *gais* | *ir* |
| Oceja (Cerdanya) | 1 | | *egibal* | *er* |
| Oceja (Cerdanya) | 1 | | *idaŕ* | *er* |
| Enveig (Cerdanya) | 1 | | *uŕmis* | *er* |
| Er (Cerdanya) | 1 | | **eŕkunbas** | *e* |
| Guils (Cerdanya) | 1 | | **anaieine** | *ka (bin)* |
| Oceja (Cerdanya) | 1 | | **unibas** | *e (IIII)* |
| L'Esquerda (Osona) | 1 | | *taŕ* | *ika* |

Of 170 Iberian rock inscriptions in Cerdanya, the corpus with which we can effectively work is around 100 inscriptions. Therefore, the repeated pattern documented in the Aquitanian inscriptions should be detected in Cerdanya with a weighted factor of approximately 3 (300/100) in the model, considering the inscriptions exclusively votive and with a factor of 4.5 (450/100) in the mixed model that considers inscriptions of different natures. In the first scenario, the number of repetitions expected in Cerdanya should be around a dozen, going down to about ten with the second one. The majority will be simple repetitions of two elements, but some can be repeated up to eight times in the first model—votive—and five in the second—mixed. In fact, the panorama that these two models predict is very similar to the one we have in Cerdanya (Table 4, with a few examples outside Cerdanya).

Aside from the combination with certain morphs, especially **e** and **er**, and the ability to appear repeatedly, several additional tendencies can be recognized. Firstly, the ending in **al** of some elements, which is not particularly frequent in Iberian inscriptions, is an interesting phenomenon that appears on many occasions. The best example of an ending in **al** is *urdal* (Figure 4), already discussed above as a very feasible theonym based on the form *\*urde*. Another good example is **tikanal**, since the base **tikan** has also been recorded as a personal name formant (Untermann 1990, no. 125; Rodríguez Ramos 2014, no. 152). Other possible formants belonging to this group would be *egibal* and **okal** (Figure 2: 13 and 20) and perhaps **edagardal** (Figure 2: 22), although the segmentation of the latter is not clear.

Another tendency of the elements that are repeated or combined with the morph **e** is that some seem to be associated with quantities. These are simple metrological expressions formed by several vertical traces that, on one occasion, accompany the word **okal** followed by the morph **e** and three units (Ferrer i Jané 2018b). Similarly, **garde** (Figure 5) uses the same pattern on four occasions, also followed by three units, and **baŕka(r)**, on one occasion, followed by the morph **ike**, and six units. It is possible that this is also the case of **anaieine**, which after the morph **ka** is followed by the expression **bin**, which in Iberian could be the number 2 (Campmajó and Untermann 1991, no. 10; Orduña 2005b; Ferrer i Jané 2009). To this group, we should add the revised text **unibas** from one inscription in Oceja (Campmajó and Untermann 1991, no. 12; Ferrer i Jané 2019). In this case, **unibas** is followed by the morph **e** and four units. It is feasible that these quantities refer to the amount of the offering/s realized by the worshipper. If that is the case, the offering made should be presupposed since there is no additional reference to the element offered to the god.

We are not yet able to clearly separate theonyms from their epithets, and therefore some of the elements identified as possible theonyms could strictly be epithets, which in some cases could function autonomously, such as Σώτειρα 'saviour', which frequently appears on anchor stocks, in reference to Ἀθηνᾶ. This could be the case of *gais*, which appears once in isolation at Sant Martí de Centelles with the morpheme **ir**. Immediately below, it appears again accompanying *aloŕberi* (*aloŕberigaisir*), which in other circumstances could be interpreted as an anthroponym and therefore identify the worshipper. However, the fact that **alorberi** also appears on a spindle whorl from Palamós (C.4.2) allows us to raise the possibility that **aloŕberi** was the theonym and *gais* its epithet. More examples are needed to confirm or reject this hypothesis.

## 4. Discussion: Peñalba de Villastar and the Iberian and Celtiberian Pantheons

Peñalba is perhaps the most significant rock sanctuary of the neighbouring ethnic group known as the Celtiberians, both because of its dimensions (over 3 km in extent) and due to the richness of its ancient inscriptions, which are currently estimated to include over twenty texts. This unique sacred place is relevant to our discussion for several reasons. First, it has traditionally been understood not only as a sacred space and as a focal point for the Celtiberian peoples but also a place of pilgrimage for the Iberians, as attested by some graffiti that a few of these pilgrims left behind (Marco and Alfayé 2008, p. 513). In the following lines, some of these texts will be reconsidered with the aim of reinterpreting certain aspects of this key sanctuary that are also relevant to suggest the existence of separate sacred landscapes and pantheons among the two groups.

First, we will evaluate if there are truly texts written in the Iberian script in Peñalba or not and if these inscriptions should be considered deities' names. Secondly, Peñalba is of interest methodologically speaking because some of the methodological approaches discussed above have been applied to this sanctuary in the past—sometimes incorrectly—to interpret its inscriptions and to identify and study Celtiberian deities. Lastly, it will be worth comparing whether the Iberians and Celtiberians shared common deities (at least as regards their names) and how many survived the Roman conquest.

### 4.1. The Supposed Iberian Inscriptions at Peñalba de Villastar

The supposed Iberian texts from Peñalba de Villastar are extremely dubious as regards their reading—since we cannot even be sure that they are epichoric Palaeohispanic scripts—and interpreting them as Iberian does not seem to be the most plausible option.

It is true that some suggested readings, such as that of Untermann, who identified **balkar** at the end of K.1.3b, would work well as a potential Iberian deity (see above), but a detailed re-examination of the inscription absolutely rules out this possibility. The rest of the signs do not show a coherent sequence of the variants of the Palaeohispanic signs. The strokes are well marked, and the surface does not seem to be damaged enough to prevent the reading to any significant extent.

In the case of K.1.2, revision of the text reveals that the strokes of the first two lines are clear, and the sequence does not work as part of an epichoric inscription. In the first line, forcing the interpretation, we could manage to read something. However, even in this case, the supposed sign **e** in Untermann's reading should be a hypothetical **m′** sign. The sign unanimously read as **l** cannot be confirmed since it is strictly speaking a vertical stroke. The same happens with the supposed sign **u**; this is insecure because it omits the diagonal right stroke, and therefore the form of the sign is not compatible with the Iberian script. Neither is the sign **te1** at the end clear since the strokes that should close the sign to the right are not apparent. The second line is even worse. The way of engraving the signs by using sinuous lines is far from the norm in Iberian inscriptions. The third line in K.1.2 is heavily damaged, and there is very little that can be said about it since only fragments of vertical strokes can be detected.

In the other two shorter inscriptions (Figure 6), it could be possible—if one wishes to do so—to force the interpretation of the signs so that an epichoric reading is feasible. However, in both cases, the resulting text will neither use the most common variants of the Iberian signs nor result in a familiar reading (for Iberian signs variants, see: Untermann 1990, pp. 246–47), especially in the case of K.1.3c, **a6** written in a way that is almost exclusively recorded in northern Catalonia and southern France. It could even work as a Latin RA, as has already been suggested. Finally, in K.1.3a, there is a quite irregular **te16**, and a levogyre (left-rotated) **r** sign in the style of the R as written in the legend of the coins of **arsaos** (BDH Mon.37), which is very rare (Ferrer i Jané and Sánchez 2017, p. 231).

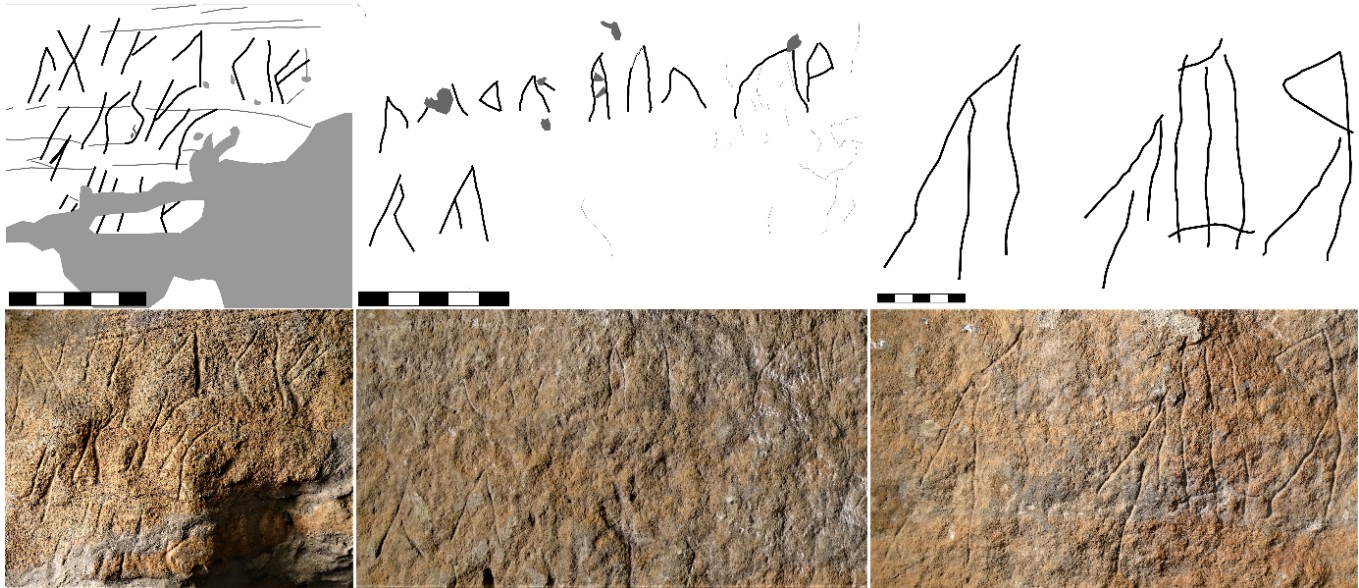

**Figure 6.** Supposed Iberian inscriptions at Peñalba de Villastar.

In sum, there are no clear Iberian inscriptions in Peñalba, and among the few possible candidates, most of their signs are strange or present aberrant forms.

### 4.2. A Shared Cultic Space for the Celtiberian and the Iberian Peoples?

If the inscriptions discussed above are not considered Iberian, the idea of Peñalba de Villastar as a 'border sanctuary' should be questioned. Nor could this cultic space be a place of pilgrimage where the nearest Iberian communities practised their religiosity, as has been traditionally argued.

Eliminating the possibility of there being Iberian dedications at Peñalba is particularly significant if we place it in a broader context together with what is known of the Celtiberian deities and the data presented in this chapter identifying Iberian theonyms. In the current state of knowledge and considering that to date, no religious space sharing dedications/texts in Iberian and Celtiberian has been identified, it seems feasible to put forward the hypothesis that the Iberian and Celtiberian sacred landscapes were dissociated. In contrast, they may have had points of contact but were, for the most part, established and separate prior to the Roman conquest. If that were the case, it would make perfect sense that even in the most important Celtiberian sanctuary known to date, located right on the border of the Iberian domains (Figure 7), not even one of the texts can be clearly associated with the peoples sharing the Iberian language.

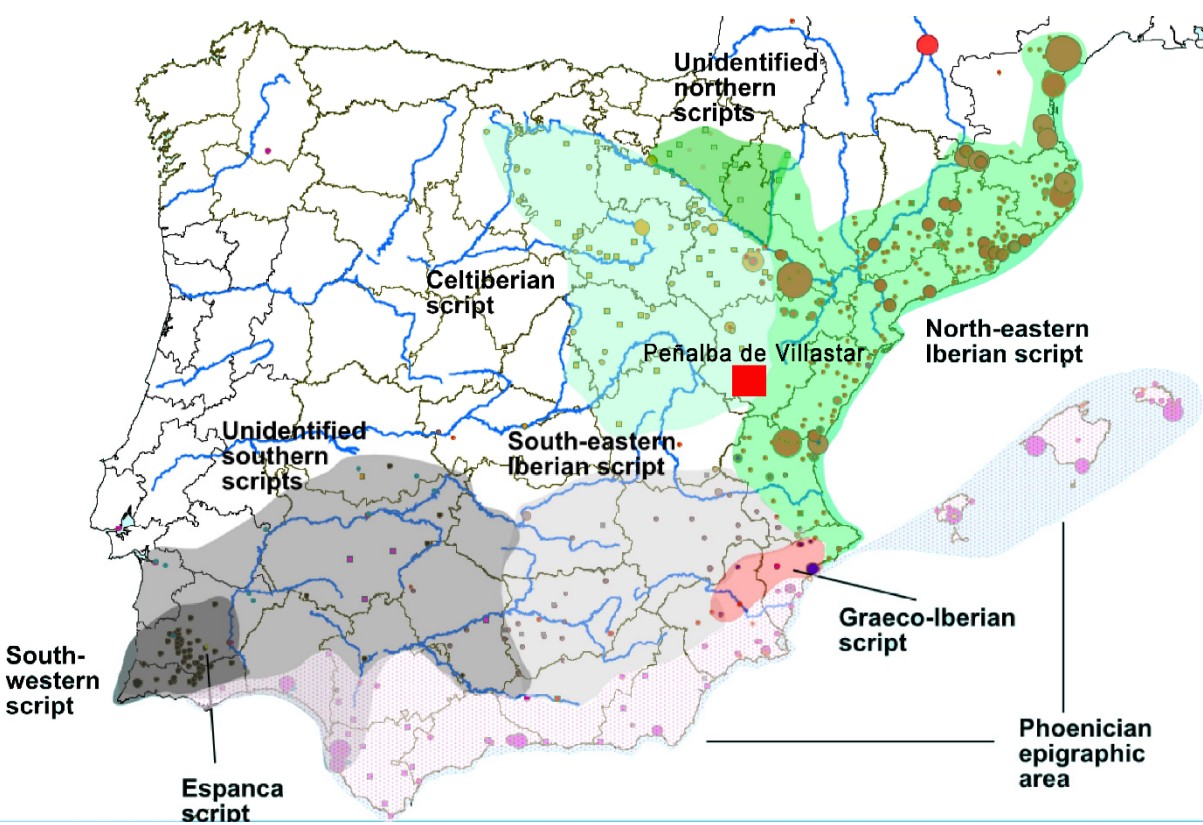

**Figure 7.** Map with the location of Peñalba de Villastar in relation to the territories where Iberian and Celtiberian inscriptions are recorded.

Additionally, the fact that none of the deity names presented above seems to be found among the Celtiberian votive dedications and vice versa points out that these two groups, if they ever shared common deities, were using different names to refer to them. Something we see, on the other hand, for example, with the Greek and Roman gods. However, the fact that so far, none of the theonyms known in Iberian and Celtiberian share similar roots seems to speak against the idea of linguistic loans between the two pantheons since, often, when deities from different cultures are integrated into each other's pantheon they retain their nomenclature (e.g., Isis).

Considering all the above, it seems feasible that a clear boundary between the two groups existed and that it was not only a territorial or linguistic one (Figure 7) but also extended to the world of beliefs. This conclusion should not be a surprise and would be expected in people that spoke languages stemming from well-differentiated linguistic families. While the Celtiberian deities have clear affinities with the divinities of Indo-European Hispania, as would be the case of *Lug*, the Iberian deities seem to have some affinities with the Basque-Aquitanian ones (Gorrochategui and Sádaba 2013); the case of the Vasconic *Vrde* and the Iberian **urdal** is perhaps the clearest. Neither should this be a surprise since all the evidence suggests that the Vasconic, Aquitanian, and Iberian languages were somehow related.

## 5. Conclusions

In the Iberian territory, from a historiographical perspective, rock sanctuaries without any built structures or associated material culture are connected to the ritual and cultural practices of the peoples and communities that inhabited the territory (Pérez Ballester 1992; Moneo 2003, pp. 308–11). On the other hand, authors such as Beard have consistently proven how the act of writing can be a key element in establishing communication and relations between an individual/collective and the deity and in helping them to constitute a divine identity (Beard 1991). The consolidation of these two elements—rock sanctuaries and writing—was fundamental in the construction of shared practices and rituals among the Iberians and helped to develop a collective memory and in the formation of sacred and symbolic landscapes.

In Indo-European Hispania, the indigenous pantheon can be reconstructed from votive Latin inscriptions that continued to invoke the same local divinities and where deities can be counted in their hundreds, some of which were both widespread and deeply rooted in the area (Velaza 2019b, p. 92). On the contrary, the votive Latin inscriptions in Iberian territory only sporadically mention three indigenous deities: *Salaeco, Sertvndo, Salagin,* and *Betatvn* (Corzo et al. 2007; Velaza 2015; Gimeno and Velaza 2021). Iberian rock inscriptions, as we have seen in these pages, allow us to start a discussion about which deities formed the Iberian pantheon since the potential deity names in rock sanctuaries now surpass twenty in number. Even if we are conservative—and we should be—just considering those that are more certain, such as the cases of **urdal**, **balkar**, **garde**, **tikanal**, **artiunan**, **gais**, **teleuś**, **okal**, and **śauś**, we can list nearly ten recorded theonyms.

The confirmation of the non-Iberian nature of the supposed Iberian inscriptions at the rock sanctuary of Peñalba de Villastar rules out the possibility of interpreting this space as a border sanctuary at which both the Iberian and Celtiberian communities worshipped. This is an important conclusion because, when added to our current knowledge—although it is still quite fragmentary—of the Iberian and Celtiberian gods, everything seems to indicate that the sacred landscapes—and perhaps even the pantheons—of these two groups were for the most part separate.

The Iberian corpus of rock inscriptions and its widespread distribution throughout the territory is the epigraphical and, in many cases, only archaeological proof of the existence of a consolidated and complex network of Iberian religious rock sanctuaries. These spaces were places of reference for the indigenous communities as well as symbolic nodes that formed a sacred landscape that was in existence prior to the Roman arrival. Most interestingly, these religious landscapes did not collapse after the Roman conquest and the adoption of the Latin language, as is proved by the survival of theonyms such as *Salaeco*, *Sertvndo*, and *Betatvn* into the Roman period. The fact that many of the rock sanctuaries discussed here later shared the space with votive and religious Latin inscriptions, the clearest case being that of El Cogul, where the text *Secvndio*/*Votvm Fecit* appears next to an Iberian text, further reinforces this hypothesis. The religious meaning of the inscriptions and symbols traced on many of the rocks studied in this chapter in later periods also points toward the religious nature of these places over the course of time.

During the Roman period, there were cases where the continuity of a previous indigenous tradition could not be proved, or if it existed, it was later completely absorbed by Roman forms and practices, sometimes even erasing all traces of the previous cult. On the other hand, the use and sometimes monumentalization of religious spaces such as Muntanya Frontera, which, as we have suggested in this study, prior to the Roman arrival, could have been dedicated to a local divinity hidden behind the forms **kauŕgo**, **kaugo**, **kaŕko,** and **kauko**(**ŕ**), seems to point to the existence of an alternative model. Here, the integration and reinterpretation of some of these ancestral religious landscapes and traditions by the local elites are a key element in understanding the transformation—or perhaps better the evolution—that the Iberian religious system suffered with the transition into the Roman period. In the case of rock sanctuaries, the process is especially important in the development and cohesion of the new rural communities. In some of the sites discussed, the divinities, rituals, and worshippers may have changed over the course of time. However, the memory that linked these symbolic spaces to a sacred landscape persisted and took part in the development of new collective identities.

**Author Contributions:** Conceptualization, A.G.S. and J.F.i.J.; methodology, A.G.S. and J.F.i.J.; software, J.F.i.J.; validation, A.G.S. and J.F.i.J.; formal analysis, J.F.i.J.; investigation, A.G.S. and J.F.i.J.; resources, A.G.S. and J.F.i.J.; data curation, J.F.i.J.; writing—original draft preparation, A.G.S. and J.F.i.J.; writing—review and editing, A.G.S. and J.F.i.J.; visualization, A.G.S. and J.F.i.J.; supervision, A.G.S.; project administration, A.G.S.; funding acquisition, A.G.S. All authors have read and agreed to the published version of the manuscript.

**Funding:** This research was funded by The Social Sciences and Humanities Research Council (SSHRC) grant number [435-2018-0777]. And by the project "Estudio paleográfico, lingüístico y funcional del corpus epigráfico ibérico (PID2019-106606GB-C33)".

**Conflicts of Interest:** The authors declare no conflict of interest.

## Abbreviations

| | |
|---|---|
| BDH | *Banco de Datos sobre Lenguas y Epigrafías Paleohispánicas:* <*http://hesperia.ucm.es/*>. |
| *CIL* | *Corpus Inscriptionum Latinarum* (Berlin, 1863–) |
| *MLH* III | *Monumenta Linguarum hispanicarum: Die iberischen Inschriften aus Spanien* (Wiesbaden: Reichert 1990) |

## Note

1 In the present text, when discussing and citing Palaeohispanic inscriptions, we have followed the conventions used and described in Sinner and Velaza (2019, p. vi).

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
