# Peer review of "Rock Sanctuaries, Sacred Landscapes, and the Making of the Iberian Pantheon"

_religions, doi:10.3390/rel13080722_

Round 1

Reviewer 1 Report

The paper shows an original content with a sound linguistic and historical background. It contributes to better understand the role of rock sanctuaries located at the boarders between Iberian and Celtiberian territories. I do recommend the publication of the article. The structure is clear and the English correct in most of the sentences. 

To further improve the content, the author may want to anticipate some of the points discussed in the conclusion and move them at the beginning of the text, to help the reader to better understand what the purpose of the article is (e.g., the author may want to anticipate the importance of rock sanctuaries for the communication between individual and divinities).

In the introduction, I would suggest to add a few general lines on the connections between the act of writing graffiti and the environment which surround the writers, a topic which has been extensively discussed for  other geographic areas e.g., in deserts.

When the author discusses the lead sheets, I would add more details and explain what their functions was.

I attach a file with minor comments and suggestions. 

These are all small changes and suggestions but the paper is already of high quality. I do appreciate the meticulous analysis of the epigraphic material.

Author Response

All the reviewer comments (in the text and the online report) have been followed/incorporated. Including an initial paragraph in the introduction anticipating the importance of rock sanctuaries for the communication between individual and divinities as well as discussing the connections between the act of writing graffiti and the environment which surround the writers has been added.

The only suggestion that has not been followed is the addition of the reference to the article Learning to Write in Indigenous Sicily A New Abecedary from the Necropolis of Manico di Quarara (Montelepre, South-West of Palermo). While the article is very interesting, in many cases (as we note in the text) abecedaries are the result of writing practices (as is the case of the suggested reference) and we have many examples of these practices from all the W med. Adding only one study would not be representative of the existing literature and adding a large number of references on a topic that is not directly related with our argument seems excessive. 

Reviewer 2 Report

This is a clearly structured and well argued paper, paying very detailed attention to the primary data of the inscriptions on the Iberian rock sanctuaries.

Some minor details/suggestions to expression and to provide additional details:

l.53: 'until a moment of time after ...' (reads oddly, reword)

l.72: 'in two different supports' (odd wording again, does the author(s) mean 'two different media'?

l.81: 'as can be seen in the stele'
Comment: the stele is not illustrated and nor is its iconographic or textual content described in sufficient detail to enable the reader to visualise/understand the relationship of the two.

l.92: problem with inconsistent capitalisation of heading

Figure 1 (map): North-eastern Iberian script (spelling error ' Nort')

l.180: 'it would also be the case of' (poor english, reword)

l.198: add a reference to the recent Bolivar publication

l.210-211: recommend adding some further detail/context to the discussion of these lead sheets. Only a reader closely familiar with the primary material would be able to understand how this relates to the suggested interpretation.

l.219: formats (not formants)

l.583: changed (not change)

Author Response

All the reviewer comments have been introduced in the revised text with the only exception of line 219 since it should say FORMANTS (forming a name) and not FORMATS